# mech2d: An Efficient Tool for High-Throughput Calculation of Mechanical Properties for Two-Dimensional Materials

**DOI:** 10.3390/molecules28114337

**Published:** 2023-05-25

**Authors:** Haidi Wang, Tao Li, Xiaofeng Liu, Weiduo Zhu, Zhao Chen, Zhongjun Li, Jinlong Yang

**Affiliations:** 1School of Physics, Hefei University of Technology, Hefei 230009, China; haidi@hfut.edu.cn (H.W.); 2021171401@mail.hfut.edu.cn (T.L.); lxf@hfut.edu.cn (X.L.); chenzhao@hfut.edu.cn (Z.C.); 2Department of Chemical Physics, and Hefei National Research Center for Physical Sciences at the Microscale, University of Science and Technology of China, Hefei 230026, China

**Keywords:** mechanical properties, 2D materials, high-throughput, DFT

## Abstract

Two-dimensional (2D) materials have been a research hot topic in the passed decades due to their unique and fascinating properties. Among them, mechanical properties play an important role in their application. However, there lacks an effective tool for high-throughput calculating, analyzing and visualizing the mechanical properties of 2D materials. In this work, we present the *mech2d* package, a highly automated toolkit for calculating and analyzing the second-order elastic constants (SOECs) tensor and relevant properties of 2D materials by considering their symmetry. In the *mech2d*, the SOECs can be fitted by both the strain–energy and stress–strain approaches, where the energy or strain can be calculated by a first-principles engine, such as VASP. As a key feature, the *mech2d* package can automatically submit and collect the tasks from a local or remote machine with robust fault-tolerant ability, making it suitable for high-throughput calculation. The present code has been validated by several common 2D materials, including graphene, black phosphorene, $GeSe_{2}$ and so on.

## 1. Introduction

The mechanical properties of 2D materials, such as graphene  [1], hexagonal boron nitride [2], transition metal dichalcogenide [3] and phosphorene [4], are crucial for determining their potential applications and performance. These materials have unique mechanical properties, such as proper strength and stiffness  [5,6], which make them suitable for a wide range of applications, including electronics [7], energy storage [8], catalyst [9], sensors [10] and magnetic devices [11]. The mechanical behavior of 2D materials can be influenced by various factors, including their structure [12], defects [13] and interfaces with other materials [14]. Accurately characterizing the mechanical properties of 2D materials is therefore essential for understanding and optimizing their performance in different applications. In particular, the second-order elastic constants (SOECs) tensor [15], which describes the material’s response to an applied strain, is a key factor in determining the mechanical properties of 2D materials.

The calculation of SOECs and the relevant mechanical properties of 2D materials via first-principles simulation is a tedious and time-consuming task which requires considering different symmetries of materials and configuring efficient computational resources [16]. Previous research has proposed algorithms and tools, such as ELASTool [17], MECHELASTIC [18], ElATools [19] and vaspkit [20], to reduce computational complexity and analyze mechanical properties. However, research gaps still exist in the field of calculating and analyzing the mechanical properties of two-dimensional materials. A clear mathematical principle for SOEC calculation of 2D materials is missing, and the presented tools lack the basic functions for task submission, monitoring, and data collection, which are crucial for high-throughput calculations [21,22]. Therefore, it is important to present mathematical principles of 2D SOECs calculation and develop an automated tool that can quickly and efficiently calculate and analyze SOECs tensor and other relevant mechanical properties of 2D materials.

In this work, we design mech2d, a highly automated toolkit for the calculation, analysis and visualization of mechanical properties of 2D materials. To be specific, mech2d allows for the calculation of SOECs using both the strain–energy approach and the stress–strain approach, utilizing first-principles engines such as the Vienna ab initio simulation package (VASP) [23,24]. The strain-energy approach (SE) concerns calculating the total energy of the system as a quadratic function of strain, while the stress–strain approach (SS) involves calculating the stress as a linear function of strain. Both approaches can be used to evaluate the mechanical properties of 2D materials. Particularly, mech2d supports automatically submitting and collecting tasks from local or remote machines, making it suitable for high-throughput calculations [21,22]. This is particularly useful for evaluating the mechanical properties of a large number of 2D materials, as it allows for efficient and robust calculations without the need for manual intervention [13]. The effectiveness of mech2d has been demonstrated based on the validation on several common 2D materials, including graphene [1], black phosphorene [4] and $GeSe_{2}$ [25] et al.

## 2. Methods

According to the Lagrangian theory of elasticity, solids can be viewed as a homogeneous and isotropic elastic medium [15]. The fundamental relationship between the physical stress tensor σ (in this work, all of the bold font letters present the matrix or vector) and physical strain tensor ϵ of the solid crystalline body within the linear regime is connected by generalized Hook’s law [15]:(1)$$\sigma_{\mu \nu }=\sum_{\lambda ,\xi \in \{x,y,z\}}C_{\mu \nu \lambda \xi }\epsilon_{\lambda \xi }$$

According to the generalized Hook law, it can be found that the physical stress tensor σ is a linear function of the physical strain tensor ϵ, where the proportionality coefficient is the forth-rank elastic tensor C. Generally, it is more convenient to use the Lagrangian stress tensor τ and Lagrangian strain tensor η; the corresponding generalized Hook law also has a similar formula:(2)$$\tau_{\mu \nu }=\sum_{\lambda ,\xi \in \{x,y,z\}}C_{\mu \nu \lambda \xi }\eta_{\lambda \xi }$$
The relationship between Lagrangian stress tensor τ and physical stress σ tensor is defined as
(3)$$\tau =det\left(I+\epsilon \right){\left(I+\epsilon \right)}^{-1}\sigma {\left(I+\epsilon \right)}^{-1}$$
where, det is the determinant of matrix, I is the 3×3 identity matrix and the physical stress tensor σ can be calculated by second-order differentiation of the total energy *E*:(4)$$\sigma =\frac{1}{V}\frac{\partial^{2}E}{\partial^{2}\epsilon }$$
where *V* is the volume of the crystal. The Lagrangian strain tensor η expression is
(5)$$\eta =\epsilon +\frac{\epsilon^{2}}{2}$$

As a center physical quantity, the elastic tensor C can be approached in many different ways. From an experimental aspect, an elastic tensor can be obtained based on sound velocity with very high precision. From a theoretical aspect, an elastic constant can be calculated by either the energy–strain or stress–strain approach [26], since most popular DFT engines can calculate energy and stress precisely. According to the Taylor’s series, the total energy *E* of a crystal can be expressed as the summation of a power series of the Lagrangian strain η:(6)$$E\left(\eta \right)=E_{0}+\frac{1}{2}V_{0}\sum_{\mu ,\nu ,\lambda ,\xi \in \{x,y,z\}}C_{\mu \nu \lambda \xi }\eta_{\mu \nu }\eta_{\lambda \xi }+\ldots$$
where $E_{0}$ and $V_{0}$ are the energy and volume of the equilibrium structure. By using the Voigt notation (xx↦1, yy↦2, zz↦3, yz↦4, xz↦5 and xy↦6), the Equation (Equation 6) can be simplified as
(7)$$E\left(\eta \right)=E_{0}+\frac{1}{2}V_{0}+\sum_{\alpha ,\beta =1}^{6}C_{\alpha \beta }\eta_{\alpha }\eta_{\beta }+\ldots$$

In a similar way, the Equation (Equation 3) can be read as
(8)$$\tau_{\alpha }=\sum_{\beta =1}^{6}C_{\alpha \beta }\eta_{\beta }$$
Therefore, under the strain–energy approach, the elastic constant $C_{\alpha \beta }$ can be expressed as
(9)$$C_{\alpha \beta }=\frac{1}{V_{0}}\frac{\partial^{2}E}{\partial \eta_{\alpha }\partial \eta_{\beta }}{|}_{\eta =0}$$
and for the stress–strain approach, the expression is
(10)$$C_{\alpha \beta }=\frac{\partial \tau_{\alpha }}{\partial \eta_{\beta }}{|}_{\eta =0}$$

As for 2D materials, we assume that the crystal lies in the xy plane; therefore, all of the elements with a subscript including *z* will be zero. To simplify the formula, we may rewrite total energy *E* and generalize Hook’s law in matrix format:(11)$$E\left(\left(\begin{array}{c} \eta_{1}\\ \eta_{2}\\ \eta_{6} \end{array}\right)\right)=E_{0}+\frac{1}{2}V_{0}\left(\begin{array}{ccc} \eta_{1} & \eta_{2} & \eta_{6} \end{array}\right)\left(\begin{array}{ccc} C_{11} & C_{12} & C_{16}\\ C_{21} & C_{22} & C_{26}\\ C_{61} & C_{62} & C_{66} \end{array}\right)\left(\begin{array}{c} \eta_{1}\\ \eta_{2}\\ \eta_{6} \end{array}\right)+\ldots$$
and
(12)$$\left(\begin{array}{c} \tau_{1}\\ \tau_{2}\\ \tau_{6} \end{array}\right)=\left(\begin{array}{ccc} C_{11} & C_{12} & C_{16}\\ C_{21} & C_{22} & C_{26}\\ C_{61} & C_{62} & C_{66} \end{array}\right)\left(\begin{array}{c} \eta_{1}\\ \eta_{2}\\ \eta_{6} \end{array}\right)$$

As can be seen, the maximum number of independent elastic constants of 2D materials has been reduced to 6, compared with their bulk counterpart of max. 21. Considering that symmetry plays an important role in elastic properties, the number of independent elastic constants of different 2D crystal structure can be further reduced according to their lattice type. Specifically, the independent elastic constants and Born stability conditions for five 2D plan Bravais (see Figure 1) lattices are listed as follow [27]:

(1)Hexagonal lattice
(13)$$C_{\alpha \beta }=\left(\begin{array}{ccc} C_{11} & C_{12} & 0\\ C_{12} & C_{11} & 0\\ 0 & 0 & \frac{C_{11}-C12}{2} \end{array}\right)$$
$$C_{11}>0\;\mathrm{and}\;C_{11}>\left|C_{12}\right|$$(2)Square lattice
(14)$$C_{\alpha \beta }=\left(\begin{array}{ccc} C_{11} & C_{12} & 0\\ C_{12} & C_{11} & 0\\ 0 & 0 & C_{66} \end{array}\right)$$
$$C_{11}>0\;\mathrm{and}\;C_{66}>0\;\mathrm{and}\;C_{11}>\left|C_{12}\right|$$(3)Rectangular and centered rectangular lattice
(15)$$C_{\alpha \beta }=\left(\begin{array}{ccc} C_{11} & C_{12} & 0\\ C_{12} & C_{22} & 0\\ 0 & 0 & C_{66} \end{array}\right)$$
$$C_{11}>0\;\mathrm{and}\;C_{66}>0\;\mathrm{and}\;C_{11}C_{22}>C_{12}^{2}$$(4)Oblique lattice
(16)$$C_{\alpha \beta }=\left(\begin{array}{ccc} C_{11} & C_{12} & C_{16}\\ C_{21} & C_{22} & C_{26}\\ C_{61} & C_{62} & C_{66} \end{array}\right)$$
$$C_{11}>0\;\mathrm{and}\;det\left(C_{\alpha \beta }\right)>0\;\mathrm{and}\;C_{11}C_{22}>C_{12}^{2}$$

After the theoretical introduction, we turn our attention to how to calculate elastic constant *C* via state-of-the-art DFT calculation. Here, we take the square lattice as an example to show how to calculate the independent elastic constant based on the energy–strain approach. It should be noted that the nature of this problem is to solve the linear equation. For the squared lattice, there are 3 independent elastic constants ($C_{11}$, $C_{12}$ and $C_{66}$), which means that we need at least 3 equations to solve this problem.

(1) **Energy–strain approach**: By substituting Equation (Equation 14) to Equation (Equation 11), the elastic energy can be written as below:(17)$$E\left(\left(\begin{array}{c} \eta_{1}\\ \eta_{2}\\ \eta_{6} \end{array}\right)\right)=E_{0}+\frac{V_{0}C_{11}\eta_{1}^{2}}{2}+\frac{V_{0}C_{11}\eta_{2}^{2}}{2}+V_{0}C_{12}\eta_{1}\eta_{2}+\frac{V_{0}C_{66}\eta_{6}^{2}}{2}$$
To simplify the above equation, a set of deformations needs to be applied. The full set of deformation types that are used in mech2d are listed in Table 1.

As for the square lattice, the required deformation set is $\eta_{1}$, $\eta_{2}$ and $\eta_{4}$. When the biaxial strain $\eta_{1}=\left(\eta ,\eta ,0\right)$ is applied, the above Equation can be simplified as
(18)$$\frac{E\left(\eta \right)-E_{0}}{V_{0}}=\left(C_{11}+C_{12}\right)\eta^{2}$$
Similarly, $C_{11}$ is obtained by using uniaxial strain $\eta_{2}=\left(\eta ,0,0\right)$:(19)$$\frac{E\left(\eta \right)-E_{0}}{V_{0}}=\frac{C_{11}}{2}\eta^{2}$$
Moreover, $C_{66}$ is calculated by using the shear strain $\eta_{4}=\left(0,0,2\eta \right)$:(20)$$\frac{E\left(\eta \right)-E_{0}}{V_{0}}=2C_{66}\eta^{2}$$

To calculate the elastic constant according to the above equation, a series of deformed structures with different Lagrangian strain (e.g., $\left[-\eta_{max},\ldots ,\eta_{max}\right]$) will be generated and evaluated by DFT engines to calculate the corresponding strain energy $\left[\Delta E_{-\eta_{max}},\ldots ,\Delta E_{\eta_{max}}\right]$. Then, the quadratic coefficients are determined by polynomial fitting of strain energy and Lagrangian strain. Generally speaking, a 4–6-order polynomial fitting with $\eta_{max}$ = 0.02–0.05 and 9–11 deformed structures is a reasonable setting [15]. Finally, the second-order elastic constants $C_{\alpha \beta }$ can be obtained by solving the system of linear equations which consist Equations (Equation 18)–(Equation 20).

One thing that deserves to be noted is that the Lagrangian strain η for 2D materials is defined as
(21)$$\eta =\left(\begin{array}{ccc} \eta_{1} & \frac{1}{2}\eta_{6} & 0\\ \frac{1}{2}\eta_{6} & \eta_{2} & 0\\ 0 & 0 & 0 \end{array}\right)$$
and the corresponding physical strain ϵ is
(22)$$\epsilon =\left(\begin{array}{ccc} \epsilon_{1} & \frac{1}{2}\epsilon_{6} & 0\\ \frac{1}{2}\epsilon_{6} & \epsilon_{2} & 0\\ 0 & 0 & 0 \end{array}\right)$$
Once the Lagrangian strain η is defined, the physical strain ϵ can be solved iteratively. Furthermore, the lattice vector of deformed structure $R^{\prime }$ can be calculated according to the physical strain ϵ and equilibrium lattice vector R by the following equation:(23)$$R^{\prime }=R\left(I+\epsilon \right)$$

So far, we have presented the energy–strain approach that is used for calculating square lattice 2D materials. For other lattice types, the corresponding deformation type can be found in Table 2.

(2) **Stress–strain approach**: In this part, we take the rectangular lattice as an example to demonstrate how to calculate the independent elastic constant based on the stress–strain approach. For the rectangular lattice, there are 4 independent elastic constants ($C_{11}$, $C_{12}$, $C_{22}$ and $C_{66}$), which means that we need at least 4 equations to solve this problem. Similar to the energy–stress approach, by substituting Equation (Equation 14) for Equation (Equation 12), the generalized Hook law can be written as below:(24)$$\left(\begin{array}{c} \tau_{1}\\ \tau_{2}\\ \tau_{6} \end{array}\right)=\left(\begin{array}{ccc} C_{11} & C_{12} & 0\\ C_{12} & C_{22} & 0\\ 0 & 0 & C_{66} \end{array}\right)\left(\begin{array}{c} \eta_{1}\\ \eta_{2}\\ \eta_{6} \end{array}\right)$$

It is obviously that for the given τ and η, 3 linear equations can be obtained according to the above matrix equation, which is not enough to calculate the 4 elastic constants. To solve this problem, at least 2 sets of deformations are needed. For the rectangular lattice, the required deformation set is $\eta_{2}$ and $\eta_{5}$. Therefore, the corresponding equation can be written as
(25)$$\left(\begin{array}{c} \tau_{2}^{\left(1\right)}\\ \tau_{2}^{\left(2\right)}\\ \tau_{2}^{\left(6\right)}\\ \tau_{5}^{\left(1\right)}\\ \tau_{5}^{\left(2\right)}\\ \tau_{5}^{\left(6\right)} \end{array}\right)=\left(\begin{array}{cccccc} C_{11} & C_{12} & 0 & 0 & 0 & 0\\ C_{12} & C_{22} & 0 & 0 & 0 & 0\\ 0 & 0 & C_{66} & 0 & 0 & 0\\ 0 & 0 & 0 & C_{11} & C_{12} & 0\\ 0 & 0 & 0 & C_{12} & C_{22} & 0\\ 0 & 0 & 0 & 0 & 0 & C_{66} \end{array}\right)\left(\begin{array}{c} \eta \\ \eta \\ 0\\ \eta \\ 0\\ 2\eta \end{array}\right)$$
Furthermore, we may rewrite the above equation as
(26)τ=HC
where
(27)$$\tau =\left(\begin{array}{c} \tau_{2}^{\left(1\right)}\\ \tau_{2}^{\left(2\right)}\\ \tau_{2}^{\left(6\right)}\\ \tau_{5}^{\left(1\right)}\\ \tau_{5}^{\left(2\right)}\\ \tau_{5}^{\left(6\right)} \end{array}\right),H=\left(\begin{array}{cccc} \eta & \eta & 0 & 0\\ 0 & \eta & \eta & 0\\ 0 & 0 & 0 & 0\\ \eta & 0 & 0 & 0\\ 0 & \eta & 0 & 0\\ 0 & 0 & 0 & 2\eta \end{array}\right)\mathrm{and}\;C=\left(\begin{array}{c} C_{11}\\ C_{12}\\ C_{22}\\ C_{66} \end{array}\right)$$
The above overdetermined equation can be solved by least square method, namely: (28)$$C={\left(H^{T}H\right)}^{-1}\left(H^{T}\tau \right)=\left(\begin{array}{c} \frac{\tau_{2}^{\left(1\right)}+2\tau_{5}^{\left(1\right)}-\tau_{5}^{\left(2\right)}}{3\eta }\\ \frac{\tau_{2}^{\left(1\right)}-\tau_{5}^{\left(1\right)}+2\tau_{5}^{\left(2\right)}}{3\eta }\\ \frac{-\tau_{2}^{\left(1\right)}\tau_{5}^{\left(1\right)}+3\tau_{2}^{\left(2\right)}-2\tau_{5}^{\left(2\right)}}{3\eta }\\ \frac{\tau_{5}^{\left(6\right)}}{3\eta } \end{array}\right)$$
Likewise, a series of deformed structures (by using Equation (Equation 23)) with different Lagrangian strain (e.g., $\left[-\eta_{max},\ldots ,\eta_{max}\right]$) will be generated and evaluated by DFT engines to give the corresponding physical stress σ and calculate the Lagrangian stress τ by using Equation (Equation 3). Then, the linear coefficients are determined by polynomial fitting of Lagrangian stress and Lagrangian strain. Finally, the second-order elastic constants $C_{\alpha \beta }$ can be obtained according to Equation (Equation 28). Similarly, the second-order elastic constants $C_{\alpha \beta }$ of other lattice type can be calculated by using the corresponding deformation type, which is listed in Table 3.

## 3. Results

### 3.1. mech2d Design

The workflow of the mech2d code is shown in Figure 2. Implemented with Python, mech2d is designed in a loosely coupled mode, which is considered easy to extend and maintain. Specifically, the workflow of mechanical properties calculation is divided into three stages, including: ***init***, ***run*** and ***post***. For the ***init*** stage, the initialization works are carried out. Firstly, the equilibrium crystal structure is read from the configuration file, such as POSCAR, cif, XSF format and so on. Then, the symmetry of the 2D Bravais lattice will be determined. Finally, the deformed structures will be generated according to the required parameters, including number of deformation, maximum strain amplitude, lattice type and calculation approach. In the ***run*** stage, all of the DFT tasks will be submitted to local or remote machine without manually writing the submitting script. As is known, fault tolerance is a key problem for high-throughput calculation. In the mech2d package, the basic errors of VASP during the mechanical properties calculation will be automatically fixed and resubmitted to the server. Once all of the DFT tasks are finished, the calculation results will be collected to working direction from a local or remote machine. In addition, the time-consuming ***run*** stage supports the task to restart. Specifically, tasks that are interrupted for external reasons will be submitted automatically, while the tasks that are already finished or still running will not be submitted. For the ***post*** stage, an internal DFT validator will be used to check the validity of results. Once passed the results checking, the elastic constant or stress–strain curve will be calculated, and the corresponding results will be output to texts and figures. Based on an object-oriented design rule, the mech2d mainly includes four classes, as detailed below.

**Elastic class**: As the core class of mech2d, the **Elastic class** is written in *mechanics.py* file. The **Elastic class** is used to initialize the mechanical properties calculation, including symmetry detection, Lagrangian strain set selection, Lagrangian strain and physical strain conversion, deformed structure generation and so on. In addition, the postprocessing of mechanical properties calculation is also employed by **Elastic class**, such as stress–strain fitting and energy–strain fitting.

**Calculation class**: This class focuses on DFT task initialization, running and result parsing. Three files are included in the calculation subfolder, including *calculator.py*, *vasp.py* and *runtask.py*. The **Calculator class** in the *calculator.py* file is the base class that defines the basic method that should be implemented in the subclass. The **VASP class** in the *vasp.py* file is a subclass of **Calculator**, which is used to prepare the VASP calculation tasks and to parse the energy and stress from the *vasprun.xml* file. As for the *runtask.py* file, it contains the **RunTasks class**. The **RunTasks class** is a wrapper of the open-source code *dpdispatcher*, which is part of our previous work [28] and is used to operate large-scale task management in machine learning potential development. As a key component, the job management framework is employed by *custodian*, an open-source code that performs error checking, job management and error recovery.

**Analysis class**: This class is used to calculate the main mechanical properties of 2D materials, such as $E_{x}$, $E_{y}$, $G_{x}y$, $\nu_{xy}$, $\nu_{yx}$, orientation-dependent Young’s modulus and Poisson’s ratio. In addition, the Born stability condition will be calculated according to the lattice type.

**Plot class**: This class is responsible for data visualization, including energy–stress fitting curve, stress–strain fitting curve, stress–strain curve under tensile strength and the polar plot of orientation-dependent Young’s modulus and Poisson’s ratio.

This main features of mech2d are listed below:(1)Easy to install (see below).(2)Loosely coupled mode, which is considered easy to extend and maintain.(3)Support for any symmetry of 2D materials.(4)Support for the popular DFT engine VASP. It can be easily extended to other DFT calculators by implementing the corresponding input writer and output parser.(5)Automatical task submission, error correction and collection on both local or remote machines.

### 3.2. Installation

Developed by python, the simplest way to install the mech2d is using *pip*. The mech2d code can be installed by downloading and decompressing the code and then running the following command “*pip install.*” in the source code directory. One thing that deserves to be noted is that three necessary libraries should first be installed, which include the following:pymatgen.dpdispatcher.custodian.

### 3.3. Running the Code

mech2d provides a user-friendly interface, which can be started via a single line of command. For example, the help information can be obtained by the following command (more details see in the Appendix A and Appendix B).


m2d -h

usage: m2d [-h] [-v] {init,run,post} …

Desctiption:
------------
mech2d is a convenient script that use to calculate the mechanical

properties of 2D materials, including Stress-Strain Curve, elastic

constants and relevant properties. The script works based on

several subcommands with their own options. To see the options

for the subcommands, type ‘‘m2d subcommand -h’’.





positional arguments:

 {init,run,post}

  init       Generating initial data for elastic systems.

  run        Run the DFT calculation for deformed

           structures.

  post       Postprocessing for elastic calculation.





optional arguments:

 -h, --help     show this help message and exit

 -v, --version   Display version


The initialization of the calculation of elastic constant calculation by using the stress–strain approach can be specified by the following command:m2d init -c POSCAR -m 0.05 -n 9 -a stress -p elc

After running the above command, the corresponding deformed structures will be generated in the *elc_stress* folder.

To run the DFT calculations, the following command can be used:m2d run -a stress input.yaml

Here, the *input.yaml* parameter specifies the input file name. In this file, the machine used to conduct the calculation, the queue system of the machine, the resources of hardware information and DFT code input information are supplied. An example of *input.yaml* is supplied in the Appendix A and Appendix B.

To postprocess the mechanical properties calculation, the following command can be used:m2d post -a stress --plot

Then the elastic constant will be calculated and orientation-dependent Young’s modulus and Poisson’s ratio will be plotted. As an example, we show the direction-dependent Young’s modulus and Poisson’s ratio of $GeSe_{2}$ in Figure 3, which is consistent with previous work [25].

### 3.4. Examples

In this section, we present some results of mechanical properties of some typical 2D materials calculated by our mech2d code. There are six test cases in this work, including graphene, $MoS_{2}$, penta-graphene, FeSe, black phosphorene and $GeSe_{2}$. The energies and stresses are calculated by the VASP software package [23,24]. The generalized gradient approximation (GGA) of Perdew, Burke and Ernzerhof (PBE) exchange correlation functional [29] and plane wave basis set are used to describe the valence electrons, with the cutoff set to 520 eV [30]. The energy convergence criteria for static calculation is set to be $1.0\times 10^{-5}$ eV. The geometry optimization is converged when the force on each atom is smaller than 1 × $10^{-2}$ eV$\cdot^{-1}$. Table 4 shows the calculation details for these cases in the present work.

Table 5 presents the in-plane elastic constants of various 2D materials, including graphene, $MoS_{2}$, penta-graphene, FeSe, alpha-phosphorene and $GeSe_{2}$. The calculation results suggest that the values for the elastic constants reported in this work are consistent with previous studies. For example, in graphene, the values for $C_{11}$ and $C_{12}$ reported in this work are in good agreement with those reported in refs. [20,25,31,32,33,34,35]. One thing that deserves to be noted is that there are slight differences in value between our work and references, which may be due to differences in simulation accuracy and methods. Overall, the values for the elastic constants reported in this work are in agreement with previous studies, validating this work.

## 4. Conclusions

In this work, we designed a python-based open-source software, mech2d, for calculating the SOECs and relevant mechanical properties of 2D materials. The package allows the calculation of SOECs with the assistant of the strain–energy approach and the stress–strain approach via first-principles engines. One essential feature of mech2d is its ability to automatically submit and collect tasks from local or remote machines, making it suitable for high-throughput calculations. This is particularly useful for evaluating the mechanical properties of a large number of 2D materials, as it allows for efficient and robust calculations without the need for manual intervention. The effectiveness of mech2d has been demonstrated through its validation on several common 2D materials, including graphene, black phosphorene, $GeSe_{2}$ et al.

The mech2d package represents a valuable resource for researchers in terms of studying the mechanical properties of 2D materials. It provides a method for calculating SOECs and other relevant mechanical properties with high efficiency, and its automation capabilities make it suitable for high-throughput calculations. This can greatly facilitate the understanding and optimization of 2D materials for a wide range of applications, such as electronics, energy storage and structural materials. Further research and development of the mech2d package may result in even greater capabilities and applications in the field of 2D material mechanics.

## Figures and Tables

**Figure 1 molecules-28-04337-f001:**
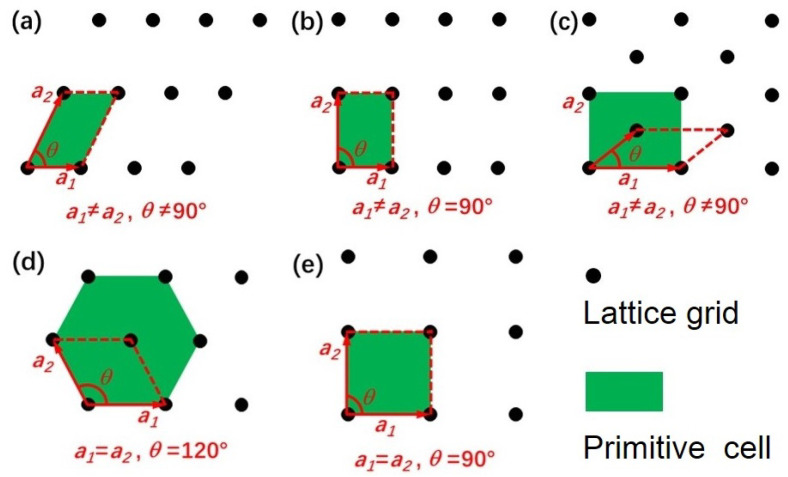
Five types of 2D Bravais lattices. (**a**–**e**) are oblique, primitive rectangular, centered rectangular, hexagonal and square, respectively.

**Figure 2 molecules-28-04337-f002:**
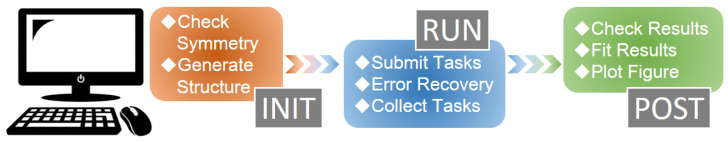
Workflow of mech2d code.

**Figure 3 molecules-28-04337-f003:**
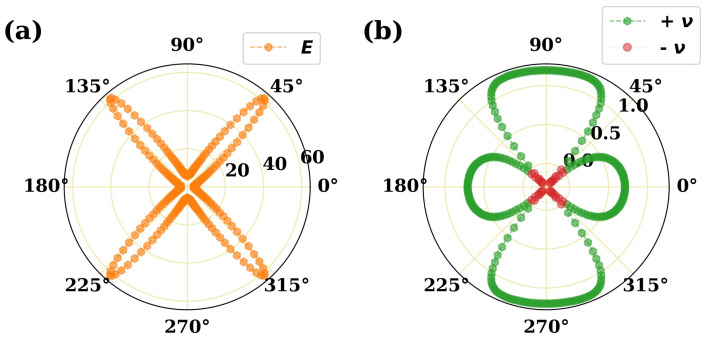
The direction-dependent Young’s modulus (**a**) and Poisson’s ratio (**b**) of 2D $GeSe_{2}$ material. Green circles denote positive values and red circles stand for negative ones for Poisson’s ratio.

**Table 1 molecules-28-04337-t001:** Deformation types that are used in mech2d with Voigt notation. The generic (*i*-th) Lagrangian strain tensor is represented as a vector: $\eta_{i}=\left(\eta_{1},\eta_{2},0,0,0,\eta_{6}\right)$. In the table, we only list $\eta_{1}$, $\eta_{2}$ and $\eta_{6}$.

$\eta_{i}$	$\eta_{1}$	$\eta_{2}$	$\eta_{6}$
$\eta_{1}$	η	0	0
$\eta_{2}$	η	η	0
$\eta_{3}$	0	η	0
$\eta_{4}$	0	0	2η
$\eta_{5}$	η	0	2η
$\eta_{6}$	0	η	2η

**Table 2 molecules-28-04337-t002:** Lagrangian strain sets used for different 2D Bravais lattices in the energy–strain approach. The deformation list is shown in Table 1.

Lattice Type	Number of Deformation	Deformation Lists
Hexagonal	2	$\eta_{1}$, $\eta_{2}$
Square	3	$\eta_{1}$, $\eta_{2}$, $\eta_{4}$
Rectangular	4	$\eta_{1}$, $\eta_{2}$, $\eta_{3}$, $\eta_{4}$
Oblique	6	$\eta_{1}$, $\eta_{2}$, $\eta_{3}$, $\eta_{4}$, $\eta_{5}$, $\eta_{6}$

**Table 3 molecules-28-04337-t003:** Lagrangian strain sets used for different 2D Bravais in the stress–strain approach. The deformation list is shown in Table 1.

Lattice Type	Number of Deformation	Deformation Lists
Hexagonal	1	$\eta_{5}$
Square	1	$\eta_{5}$
Rectangular	2	$\eta_{2}$, $\eta_{5}$
Oblique	3	$\eta_{1}$, $\eta_{2}$, $\eta_{5}$

**Table 4 molecules-28-04337-t004:** The lattice type, lattice parameters and k-points for 2D test cases.

System	Lattice Type	a (Å)	b (Å)	γ(°)	K-Points
graphene	Hexagonal	2.468	2.468	120	24 × 24 × 1
$MoS_{2}$	Hexagonal	3.180	3.180	120	21 × 21 × 1
penta-graphene	Square	3.631	3.631	90	17 × 17 × 1
FeSe	Square	3.671	3.671	90	18 × 18 × 1
Phosphorene	Rectangular	3.297	3.297	90	25 × 15 × 1
$GeSe_{2}$	Centered-Rectangular	4.906	5.571	90	15 × 12 × 1

**Table 5 molecules-28-04337-t005:** The 2D in-plane elastic constants (N/m) of 2D systems in comparison with references. SS stands for stress–strain approach and ES stands for energy–strain approach.

System	Sources	Approach	$C_{11}$	$C_{12}$	$C_{22}$	$C_{66}$
graphene	our work	ES	354.1	67.5		
	our work	SS	353.2	63.1		
	Ref. [20]	ES	349.1	60.3		
	Ref. [31]	ES	358.1	60.4		
	Ref. [32]	SS	353.2	63.7		
$MoS_{2}$	our work	ES	132.6	32.8		
	our work	SS	134.4	34.6		
	Ref. [32]	SS	136.9	33.1		
	Ref. [33]	SS	131.4	32.6		
	Ref. [20]	ES	128.9	32.6		
penta-graphene	our work	ES	269.5	−20.1		151.4
	our work	SS	270.1	−18.6		151.4
	Ref. [35]	ES	265.0	−18.0		/
FeSe	our work	ES	58.4	20.5		38.3
	our work	SS	62.4	25.7		36.9
	Ref. [32]	SS	58.2	22.7		38.1
α-phosphorene	our work	ES	103.8	17.1	24.4	22.7
	our work	SS	103.4	17.8	23.9	22.6
	Ref. [34]	ES	105.2	18.4	26.2	22.4
	Ref. [20]	ES	104.4	21.6	34.0	27.4
	Ref. [32]	SS	103.4	18.0	24.6	21.8
$GeSe_{2}$	our work	ES	23.2	28.1	39.4	31.6
	our work	SS	23.7	28.3	39.4	31.5
	Ref. [25]	ES	23.0	27.4	37.8	31.1

## Data Availability

The code can be found at https://gitee.com/haidi-hfut/mech2d (accessed on 20 May 2023).

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
