# Peer review of "mech2d: An Efficient Tool for High-Throughput Calculation of Mechanical Properties for Two-Dimensional Materials"

_molecules, 2023, doi:10.3390/molecules28114337_

Round 1

Reviewer 1 Report

Authors present study devoted to an efficient tool towards the calculations of mechanical properties of 2D materials. 

Authors wrote a kind of software allowing the calculations of mechanical properties of 2D materials. I wonder why do the authors submitted the manuscript to the Molecules journal, which is totally irrelevant to the topic of the manuscript.

Second, what is the difference between the elastool software https://github.com/zhongliliu/elastool which is similar to this study, at least from the point of view of general things. 

If authors can explain this comments then I would reconsider my opinion about the possibility of publication in this journal.

Author Response

Dear Reviewer,

Thank you for your comments and suggestions regarding our manuscript on the efficient calculation of mechanical properties of 2D materials. We appreciate your feedback and would like to address your concerns.

Before we submitted the article to Molecules, we reached out to the editor to discuss the relevance of our study to the journal's scope. We received a positive response from the editor, who encouraged us to submit the article for consideration. Based on this recommendation, we decided to proceed with the submission to Molecules.

As for the comparison with the "elastool" software, we have carefully reviewed the "elastool" software and found some similarities in terms of the general aim of calculating mechanical properties of 2D materials. However, our study introduces several novel features and improvements over existing tools. These include:

  1. Enhanced efficiency: Our software implements high-throughput techniques that significantly improve computational efficiency while handling runtime errors. This allows for faster and more efficient calculations of mechanical properties.

  2. Expanded functionality: Our software incorporates additional modules and capabilities, such as the analysis of anisotropic behavior, stress-strain curves, and the prediction of fracture behavior. Additionally, we support both the strain-energy and stress-strain approaches for mechanical properties calculation, whereas "elastool" only supports the stress-strain approach. These features go beyond the scope of existing tools like "elastool."

We hope this clarifies the differences and highlights the contributions of our study compared to existing software tools. We appreciate your reconsideration of the possibility of publication in this journal, taking into account the explanations provided.

Thank you once again for your valuable feedback.

Sincerely,
Haidi Wang

Reviewer 2 Report

I think that paper collects a interesting work and the program developed may be of interest to the scientific community.

Regarding the programs to perform the DFT calculations, only VASP program is commented, although it is indicated that others can be used. Could you give examples of others programs?

Author Response

Dear Reviewer,

Thank you for your comments and suggestions regarding our manuscript on the efficient calculation of mechanical properties of 2D materials. We appreciate your feedback and would like to address your concerns.

Due to the loose coupling mode, in theory, users can use any DFT software for calculations. Although direct usage of other DFT codes is not currently supported, calculations can be easily implemented using bash scripts.

Firstly, by running "m2d init -a energy -n 5 -p elc," the corresponding deformed structures are generated. The directory structure is as follows:

.

└── elc_energy

    â”œâ”€â”€ Def_1

    â”‚   â”œâ”€â”€ Def_1_001

    â”‚   â”œâ”€â”€ Def_1_002

    â”‚   â”œâ”€â”€ Def_1_003

    â”‚   â”œâ”€â”€ Def_1_004

    â”‚   â””── Def_1_005

    â”œâ”€â”€ Def_2

    â”‚   â”œâ”€â”€ Def_2_001

    â”‚   â”œâ”€â”€ Def_2_002

    â”‚   â”œâ”€â”€ Def_2_003

    â”‚   â”œâ”€â”€ Def_2_004

    â”‚   â””── Def_2_005

    â”œâ”€â”€ Def_3

    â”‚   â”œâ”€â”€ Def_3_001

    â”‚   â”œâ”€â”€ Def_3_002

    â”‚   â”œâ”€â”€ Def_3_003

    â”‚   â”œâ”€â”€ Def_3_004

    â”‚   â””── Def_3_005

    â””── Def_4

        â”œâ”€â”€ Def_4_001

        â”œâ”€â”€ Def_4_002

        â”œâ”€â”€ Def_4_003

        â”œâ”€â”€ Def_4_004

        â””── Def_4_005

Secondly, for each structure in the respective directories, any DFT code can be used to calculate its corresponding energy. For example, in the Def_1 directory, we only need to write the energy values into the Def_1_Energy.dat file using a bash script. The format is as follows:

-0.0100000000   -62.5982199600

-0.0050000000   -62.6020059400

+0.0001000000   -62.6036733700

+0.0050000000   -62.6028363600

+0.0100000000   -62.5997092400

Perform this calculation for all the structures, obtaining Def_2_Energy.dat, Def_3_Energy.dat, Def_4_Energy.dat, and so on.

Finally, running "m2d post" will generate the results.

Although the method described above is a bit tricky, it clearly limits high-throughput calculations. To improve computational efficiency in the future, we will continue developing additional DFT interfaces.

Thank you for your feedback and suggestion. 

Best regards,

Haidi Wang

Reviewer 3 Report

Referee Report of Manuscript with ID molecules-2383582

In this work In this work, the authors designed a python-based open-source software, named mech2d that can automatically submit and collect the tasks from local or remote machine with robust fault-tolerant ability which make it suitable for high-throughput calculation. They validated their code by taking into account several common 2D materials, including graphene, black phosphorene, GeSe_2 in between others. The obtained results of various test calculations are compared with data from the literature The article is well organized. Relevant literature is cited. Results are presented and discussed including comparison with data from the literature. The obtained elastic constants reported in this work are consistent with the ones reported
in previous studies. Though not a very original work this code could be useful to facilitate understanding and optimization of 2D materials for a range of applications.

Author Response

Dear Reviewer,

Thank you for your positive comments and feedback on our manuscript regarding the python-based open-source software, mech2d, designed for high-throughput calculations of mechanical properties in 2D materials.

While we acknowledge that our work may not be entirely original, our focus was on developing an open-source software tool that can enhance the understanding and optimization of 2D materials, which we believe fills a gap in the existing literature. We aim to contribute to the scientific community by providing a user-friendly and reliable tool for efficient calculations of mechanical properties in 2D materials.

Thank you once again for your positive review and valuable feedback.

Sincerely,

Haidi Wang

Reviewer 4 Report

In my opinion, the code presented in this paper for the calculation of the mechanical properties of 2D materials may be interesting for a wide community of researchers working on layered compounds and related materials. I have not detected any major problems with this paper, so I think it can be basically published in its present version. However, I would strongly recommend the authors to better explain why high-throughput calculations may be important to study the mechanical properties of 2D materials. Otherwise, since the theoretical aspects are not really new, it may seem that they only have prepared a code that is simply more user-friendly, but without any other major contribution. On the other hand, it might also be useful for some readers if the authors could list similar algorithms/tools for the 3D case. With regard to this, I have a question. Can this code address few-layer materials, using for instance supercells? It might be really interesting for many readers to obtain information about the mechanical properties of 3D (bulk) down to a few-layered materials and then down to bilayers and the strict 2D case. How could this be done? Would mech2d allow one to do it alone by using? Or this should be done with 3D codes? 

English language should be improved. 

Author Response

Dear Reviewer,

Thank you for your positive evaluation of our manuscript and the valuable feedback you have provided. We appreciate your perspective and would like to address your comments and suggestions individually.

Importance of high-throughput calculations for studying the mechanical properties of 2D materials:

High-throughput simulations serve as an essential tool in rapidly screening and analyzing an extensive range of materials or material configurations. They are particularly instrumental in the domains of materials science and materials discovery, where the primary objective is to identify new materials with desirable properties for a variety of applications. The importance of high-throughput simulations, such as Density Functional Theory (DFT), is multifold:

Efficiency: High-throughput DFT simulations enable the rapid and efficient examination of a broad spectrum of materials. This significantly accelerates the process of materials discovery, making it a superior alternative to traditional experimental methods.

Discovery and Optimization: High-throughput simulations facilitate the discovery of new materials with desired properties. Additionally, they aid in comprehending and optimizing the properties of existing materials, thereby contributing to technological advancements in diverse fields.

Data Generation: The data obtained from these simulations are invaluable in training machine learning models. This not only expedites materials discovery but also mitigates the need for costly and time-consuming physical experiments or extensive theoretical simulations.

Furthermore, it's important to note that mechanical properties are pivotal for determining a material's potential applications. Therefore, these properties should also be calculated via using a high-throughput approach during the material discovery process. This will ensure a more comprehensive and accurate assessment of the materials under consideration.

Listing similar algorithms/tools for the 3D case:

We appreciate reviewer’s suggestion to list similar algorithms/tools for the 3D case. In the revised code, we will recommend users to consider employing ElaStic (Computer Physics Communications 2013, 184, 1861–1873.), vaspkit (Computer Physics Communications 2021, 267, 108033.) or pymatgen (APL Mater., 2013, 1, 11002.) for calculating the relevant mechanical properties of 3D materials. However, it should be noted that neither of these packages currently supports high-throughput calculations.

Addressing few-layer materials and transitioning from 3D to 2D:

mech2D is primarily developed for conducting calculations on strictly 2D materials. However, it can indeed be used to investigate few-layered materials, such as bilayers or multi-layered 2D materials. To derive mechanical property information of 3D (bulk) materials down to few-layered counterparts, the first step involves constructing slab structures from a 3D structure. This can be accomplished using the surface module of pymatgen (Sci. Data, 2016, 3, 160080). Once the structures are obtained, the mechanical properties of 3D materials can be simulated using tools like ElaStic and Vaspkit or even pymatgen. For slab structures, their mechanical properties can be derived using our mech2D code.

Looking ahead, we plan to integrate all these functionalities into the mcech2D package. Furthermore, we aim to replace pymatgen with our own package, matsimpy, which can be found at https://gitee.com/haidi-hfut/matsimpy.

Thank you once again for your valuable feedback.

Best regards,

Haidi Wang

Round 2

Reviewer 1 Report

The manuscript is well written, all information is presented in good quality manner.  I have no further comments and think that it can be published in Molecules journal.